# Coping Strategies of Smallholder Coffee Farmers under the COVID-19 Impact in Indonesia

Suci Wulandari [1,*], Fadjry Djufry [2] and Renato Villano [3]

1 Indonesian Center for Estate Crops Research and Development, Bogor 16111, Indonesia
2 Indonesian Agency for Agricultural Research and Development, Jakarta 12540, Indonesia; fadrydjufry@pertanian.go.id
3 UNE Business School, University of New England, Armidale, NSW 2351, Australia; rvillan2@une.edu.au
* Correspondence: suciwulandari@pertanian.go.id

**Abstract:** COVID-19 significantly impacts coffee production, which smallholders dominate. Un-addressed impacts will affect coffee production sustainability. However, smallholders face some constraints. This study aims to determine the impact of COVID-19 from the perspective of Arabica and Robusta farmers in Indonesia, examine technical recommendations as coping strategies, and develop an institutional model to accelerate implementation. We analyzed the divergences in the perceptions of different categories and clusters using farm-level data. Descriptive statistics, Mann–Whitney analysis, cluster analysis, and crosstab analysis were used to explore the facts. Immediate effects of COVID-19 were observed through a decline in household income, impacting the allocation of farming costs, which influences productivity related to the intensity of cultivation, particularly in purchasing and fertilization decisions. It was explored by the study that coffee livestock integration is an important strategy to improve farmers' livelihood to mitigate the impact. The innovation sharing model complements this technical recommendation as institutional recommendations, including innovation sharing elements and processes. Four farmer clusters have been identified based on the adoption spectrum and farmer conditions. The intervention provides innovation-sharing elements for farmers who have not adopted integration. Where integration was partially completed, reusing waste is recommended by completing innovation elements and improving the sharing process.

**Keywords:** productivity; coffee; smallholders; integrated farming; innovation sharing





## 1. Introduction

COVID-19 significantly impacts the global coffee sector, including production, consumption, and international trade [1]. The decrease in consumer income is expected to reduce the demand for coffee [2]. According to the ICO, a one percentage point reduction in GDP growth is associated with decreased global demand growth for coffee by 0.95 percentage points or 1.6 million 60 kg bags [3]. The demand for coffee is extremely high. Global consumption has fallen only twice since 2000.

Although COVID-19 is likely to have little impact on long-term demand, the pandemic is predicted to impact coffee smallholders significantly, especially in production [4]. While COVID-19 is unlikely to have a long-term effect on the market, the pandemic is causing problems for smallholders. Smallholders are characterized by poor management of the coffee farming system [5,6] and unpreparedness to deal with shocks [7]. Additionally, the limited adoption of technology is determined by the availability of off-farm income, access to credit, adequate land size, and confidence in land ownership [8]. Establishing the resilience of the coffee sector is predicted to be slow [9].

Both farm households and farming systems have felt the pandemic's influence. Coffee farmers' lives are under increased pressure as agricultural revenues decline while input costs and expenditures on food and non-food products increase [1]. The threats that

COVID-19 pose to the global coffee sector are daunting, with profound implications for coffee production [4]. In the context of vulnerability due to the existing coffee production system, the socioeconomic disruption due to COVID-19 is expected to cause the coffee sector to experience another severe production crisis [10]. The implementation of social distancing and regional restrictions will also impact production related to the availability of labor and input production [11]. The financial impacts will be long-lived and uneven, and smallholders will be among the hardest hit. For coffee farmers, this is exacerbated by conditions rooted in the systemic vulnerability of the coffee production system [4].

Several recent studies have attempted to analyze the impact of COVID-19 on coffee agribusiness. The survey of ICO exporting members reveals that countries have experienced a largely negative impact of COVID-19 in the coffee sector and that the outlook remains uncertain [1]. The influence of COVID-19 on the coffee farming system was also examined, although only in a descriptive manner, on the economic, social, and institutional levels [10]. Additionally, an impact analysis was conducted on coffee shops in response to COVID-19, and adjustments to the marketing plan were undertaken. These numerous researches, however, are generic in design. On the other hand, the impact of COVID-19 differs across agricultural systems; therefore, the strategies used and the results achieved by these strategies also vary across systems [10]. COVID-19′s impact on the coffee sector varies by country due to the epidemic curve, the coffee harvest cycle, and the production system [1]. The generic analysis will produce generic recommendations, which are operationally difficult to implement. Cultivated coffee in Indonesia is classified as Arabica coffee (*Coffea arabica*) and Robusta coffee (*Coffea canephora*). This study will provide a detailed description of the impact of COVID-19 on the Robusta and Arabica coffee farming systems.

Due to the perennial nature of coffee and its delayed pandemic effect, evaluating the pandemic's impact on smallholders is challenging. As a result, an evaluation methodology based on farmer perceptions could be one method for forecasting the impact of COVID-19. Farmers' behavioral patterns and decision-making tendencies are the foundation for a more technical impact analysis. Additionally, technical recommendations cannot be applied automatically due to limitations at the farmer level. Therefore, technical recommendations need to be followed by institutional recommendations to encourage smallholders to adopt technology [8,12]. With increasing farm-level technology adoption, technical recommendations are insufficient; they must be complemented by an inclusive innovation process in agricultural systems [13]. This study will include technical advice for coping with the impact of COVID-19 and institutional recommendations to support farmer adoptions.

Among the many commodities produced in Indonesia, coffee is one of the most important, contributing 16.15% to the GDP [14]. Coffee exports are ranked as the third-largest plantation commodity in Indonesia after oil palm and rubber. Smallholders dominate 96.63% of coffee plantations in Indonesia [15]. Robusta coffee accounts for 88.93% of production, and Arabica coffee accounts for the remainder. Therefore, this research will be useful in improving the national economy and supporting small coffee farmers in being more resilient.

This study aims to determine how COVID-19 has impacted Arabica and Robusta farmers in Indonesia. The following research questions are addressed in this study: (1) what are the perceptions of Arabica and Robusta farmers in Indonesia regarding the impact of COVID-19, (2) what coping strategies should be formulated, and (3) what institutional model should be developed? Mapping COVID-19′s impact on Arabica and Robusta farmers, compiling technical recommendations as a coping strategy, and developing institutional models supporting its implementation will significantly contribute to the coffee sector that smallholders dominate.

This study is expected to provide an overview of the COVID-19 impact on farmer households and the coffee farming system and provide a framework for developing a strategy for limiting the impact of COVID-19. With modest modifications, the findings of this study can also be applied to coffee-producing systems in other locations. It is believed

that a more resilient smallholder plantation-based production system in the context of sustainable coffee production will be established by implementing the proposed strategy.

## 2. Materials and Methods

This study evaluated farmer perceptions through a multipronged method. Farm-level data was collected from the study area to conduct a quantitative analysis in profiling and clustering farmers. Accordingly, the cluster analysis results target different farmers and their respective coping strategies. Details of the study area and methods are provided below.

### 2.1. Study Area and Respondent's Selection

In Indonesia, Arabica coffee production centers include the provinces of West Java, North Sumatra, Aceh, South Sulawesi, and West Sumatra. Meanwhile, Robusta coffee is grown in Bengkulu Province and South Sumatra, Lampung, East Java, and Central Java. Two areas were purposely selected based on two distinct coffee production centers: Bandung Regency in West Java Province for Arabica farmers and Rejang Lebong and Kepahiang Regencies in Bengkulu Province for Robusta farmers (Figure 1).

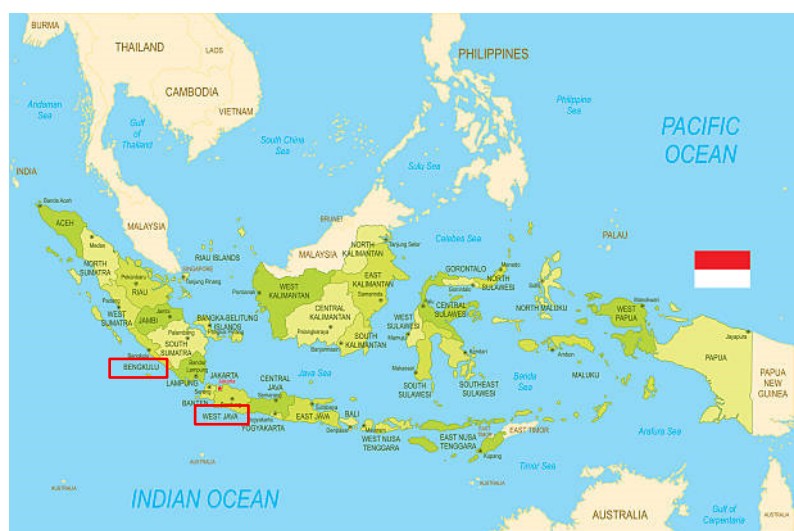

**Figure 1.** Survey locations.

Bandung Regency is a mountainous region with an average slope of 0–8%, increasing to 8–15% and exceeding 45%. The average monthly rainfall is 266 mm. The air temperature ranges between 12 and 24 °C, with humidity levels ranging between 78 and 70% during the rainy season and 70% during the dry season. Rejang Lebong Regency is a hilly region with slopes ranging from flat to wavy. The average monthly rainfall is 233.75 mm. The typical normal temperature is between 18 and 31 °C, with an average humidity of 85.5%. Kepahiang Regency is located in a mountainous region. The average monthly rainfall is 233.5 mm, with an average humidity of 85.21% and an average daily temperature of 23.87 °C, ranging from 29.87 °C to 19.65 °C.

The study was conducted in November 2020. Data was collected through interviews with 205 selected farmers with productive coffee plants. The number of respondents was 205, consisting of 79 respondents from Arabica farmers in West Java Province and 156 respondents from Robusta farmers in Bengkulu Province.

### 2.2. Methods of Analysis and Data

A multistage approach was applied to analyze the survey data using descriptive statistics, Mann–Whitney analysis, cluster analysis, and crosstab analysis. Firstly, using the Mann–Whitney test, we examined the differences in the perceptions of Arabica and Robusta farmers about the impact of COVID-19 on farmer households and coffee farming systems.

Secondly, we used cluster analysis to categorize similar observations into groups based on the values of the observations related to several strategies at the farm level to cope with the impact of the pandemic. Thirdly, we used cross-tabulation to analyze farmer profiles and classify them into different groups with certain similarities. Lastly, we developed a conceptual model based on a smallholder innovation strategy that aims to facilitate the implementation of technical recommendations. The analysis path is depicted in Figure 2.

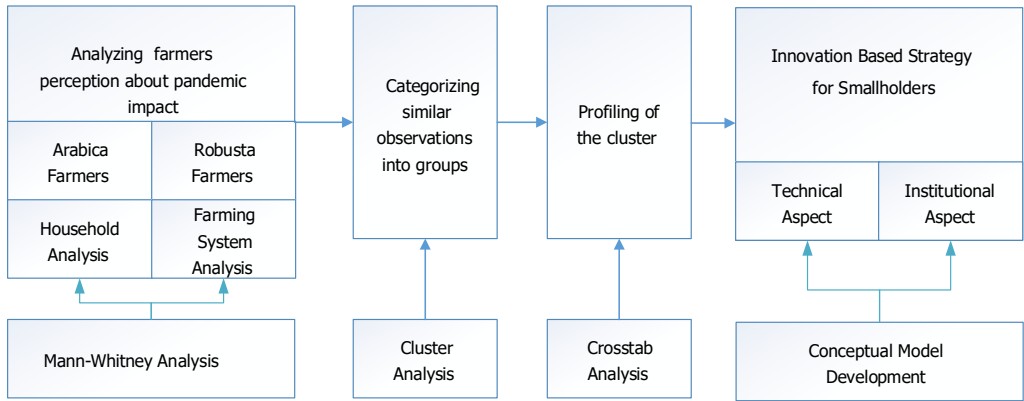

**Figure 2.** Analysis stages and method of analysis.

The analysis was carried out on farmer households and coffee farming by exploring a number of variables (Table 1). The socioeconomic impact of COVID-19 on individuals is predicted by developing a microeconomic model by analyzing the household income, savings, consumption, and poverty [16]. In addition, the impact of COVID-19 can be explained by several factors in the food system, such as food price [17]. The pandemic has also had a significant impact on internet usage [18], and predictions will be followed by an increase in electricity use. By considering the condition of coffee farmers in the field, the factors analyzed in this study included income, expenditure, consumption, food quality, food price, food type, internet cost, and electricity cost.

**Table 1.** Variables of analysis.

| Aspect | | Variables | Description |
|---|---|---|---|
| Farmers household | | Income | Income earned |
| | | Expenditure | Expenditures spent |
| | | Consumption | Household food consumption |
| | | Food Quality | Food quality |
| | | Food Price | Food prices |
| | | Food Type | Types of food consumed |
| | | Internet Cost | Cost of expenses for internet service |
| | | Electricity Cost | Electricity expenses |
| Coffee farming system | Input | Fertilizer | Availability of chemical fertilizers |
| | | Organic Fertilizer | Availability of organic fertilizers |
| | | Pesticide | Availability of pesticides |
| | | Labor | The use of external labor |
| | | Equipment | Access to coffee processing equipment |
| | | Capital | Availability of working capital |
| | Process | Cultivation | Application of Good Agricultural Practices (GAP) |
| | | Pest Disease Management | Management of pests and diseases in coffee plants |
| | | Harvesting | Harvesting technique |
| | | Marketing channel | Marketing channels in product sales |
| | | Transaction method | Transaction method in marketing |

An analysis of the impact of COVID-19 on farm operations can be viewed from land, labor, inputs, production, and markets [19]. The input aspects discussed in this study are the main agricultural inputs used in coffee production, including fertilizer, organic fertilizer, pesticide, labor, equipment, and capital. The production aspect or process aspect can be divided by the pre-harvest and post-harvest levels [20], including cultivation, pest diseases management, harvesting, and marketing. In the case of small-scale coffee farmers, the relevant marketing aspects include marketing channels and transaction methods.

The impact on farmer households is calculated using a scale of increase, no change, or decline. Measurements of the impact analysis on the farming system used a Likert Scale with the levels of influence consisting of 1—not at all influential, 2—slightly influential, 3—somewhat influential, 4—very influential, and 5—extremely influential.

The Mann–Whitney test's null hypothesis ($H_0$) was established that the two independent groups are homogeneous and have the same distribution [21]. If the significance value >0.05, then $H_0$ is accepted, and $H_1$ is rejected. If the significance value <0.05, then $H_0$ is rejected, and $H_1$ is accepted.

$$U_i = n_1 n_2 + \frac{n_i(n_i + 1)}{2} - \sum R_i \qquad (1)$$

Description:

U = Statistic test of $U_1$
$R_i$ = number of rank sample i
$n_1$ = number of samples 1
$n_2$ = number of samples 2

A cluster analysis is a technique for categorizing similar observations into groups based on the observed values of numerous variables for each individual. This study used the K-means cluster method, in which the centroid is made up of the mean values of the individual variables, called k-means. ANOVA was used to analyze the differences between the variables in each cluster. The formula for the F value is:

$$F = \frac{\text{between Means}}{\text{within Means}} \qquad (2)$$

The more the F-number of a variable when the significance level is less than 0.05, the greater the difference between variables within the created cluster.

## 3. Results and Discussion

Since the government reported the first Corona infection in Indonesia in March 2020, the COVID-19 epidemic has plagued the country. COVID-19 has caused a public health disaster and significantly impacted national economic activities. Since April 2020, large-scale social restrictions have significantly influenced the production and distribution processes and other operational activities, disrupting the economic performance. The economic recession almost paralyzed all economic activity, both from the supply and demand sides. The agricultural sector is one of the business sectors that can withstand a contraction in economic growth.

### 3.1. Household Characteristics and Impact of COVID-19

According to the HeadCount Index, which determines the percentage of the population categorized as poor, the percentage of poor people in the two provinces where the sample was taken reflects a wide range of conditions. The poverty rate in Bengkulu, which is devoted to Robusta farmers, was 15.30% or greater than the national average (10.14%), while the rate in West Java, which is related to Arabica farmers, was lower (8.43%) [22].

The survey results on the household characteristic of Arabica and Robusta farmers can be seen in Table 2. Differences in the characteristics can be seen in education and cooperative membership. From the level of education, there is a significant difference,

where most of the respondents graduated from junior high school for Arabica farmers (38%) and senior high school for Robusta farmers (54%). From the institutional aspect, the proportion of Arabica farmers who are members of the cooperative is higher (15.2%), compared to 6.3% of Robusta farmers. Numerous cooperatives in West Java purchase and process coffee beans from farmers, thereby attracting farmers to join.

**Table 2.** Arabica and Robusta farmers' household characteristics.

| Aspects | Minimum | Maximum | Mean | Std. Deviation | Description |
|---|---|---|---|---|---|
| **Arabica Farmers** | | | | | |
| age | 20 | 75 | 48.44 | 13.003 | |
| education | 1 | 5 | 2.03 | 0.947 | 1 = Elementary (30.4%), 2 = Junior High (48.1%), 3 = Senior High (11.4%), 4 = Diploma (8.9%), 5 = University (1.3%) |
| nonagricultural activities | 0 | 1 | 0.27 | 0.445 | 1 = Yes (26.6%), 0 = No (73.4%) |
| farmers group | 1.00 | 1.00 | 1.000 | 0.000 | 1 = Yes (100%), 0 = No (0%) |
| cooperatives | 0.00 | 1.00 | 0.152 | 0.361 | Yes (15.2%), No = 84.8% |
| **Robusta Farmers** | | | | | |
| age | 20 | 73 | 40.77 | 9.655 | |
| education | 1 | 5 | 2.35 | 1.007 | 1 = Elementary (24.6%), 2 = Junior High (26.2%), 3 = Senior High (42.9%), 4 = Diploma (2.4%), 5 = University (4.0%) |
| nonagricultural activities | 0 | 1 | 0.28 | 0.450 | 1 = Yes (27.8%), 0 = No (72.2%) |
| farmers group | 1.00 | 1.00 | 1.000 | 0.000 | 1 = Yes (100%). 0 = No (0%) |
| cooperatives | 0.00 | 1.00 | 0.0635 | 0.245 | 1 = Yes (6.3%), 0 = No (93.7%) |

Based on age, the average age of respondent Arabica farmers is 48.44 years, while Robusta farmers are 40.77 years. The youngest Arabica and Robusta farmer is 20 years old, while the oldest is 75 years for the Arabica farmers and 73 years old for the Robusta farmers.

Most respondents did not engage in nonagricultural activities, whether Arabica (73.4%) or Robusta farmers (72.2%). In West Java, coffee is the fourth main commodity after coconut, tea, and rubber. The increase in the coffee area is very significant. In 2019, the coffee plantation area was 46,125 ha, so, in 2020, it reached 81,273 ha with the dominance of Arabica coffee (98.3%).

COVID-19 has some impacts on farmer households. The majority of the rural population works in agriculture, where farming methods in rural communities are still relatively traditional, with most products still allocated for daily needs [23]. COVID-19 has exacerbated the poverty and food and nutrition insecurity in farmer households. It has affected the food demand, which then affects food security, with the most significant impact being felt by the most vulnerable populations [24]. The COVID-19 pandemic also impacts food security and the stability of food supply systems in developing countries [25]. In this study, the impact on farmer households is seen from changes in income, expenditure, consumption, food quality, food product prices, types of food, internet costs, and electricity costs.

The study reveals that Arabica and Robusta farmers' perceptions of the COVID-19 impact on Arabica and Robusta farmers' households are quite different. Expenditures, food quality, food prices, and electricity costs differ significantly (Table 3). The number of farmers in each assessment varies, which is assumed to be determined by environmental circumstances and the farming system's resilience, influencing farmers' incomes. These external and internal conditions cause differences in Arabica and Robusta farmers' perceptions regarding the impact of the pandemic.

**Table 3.** Mann–Whitney analysis for the COVID-19 impact on Arabica and Robusta farmers' households.

|  | Income | Expenditure | Consumption | Food Quality | Food Price | Food Type | Internet | Electricity Cost |
|---|---|---|---|---|---|---|---|---|
| Mann–Whitney $U$ | 4512.000 | 3768.500 | 4904.500 | 4218.000 | 2667.000 | 4597.500 | 4314.500 | 3131.500 |
| Wilcoxon W | 12,513.000 | 11,769.500 | 12,905.500 | 7378.000 | 10,668.000 | 12,598.500 | 7474.500 | 6291.500 |
| Z | −1.299 | −3.599 | −0.242 | −2.181 | −6.528 | −1.475 | −1.860 | −5.143 |
| Asymp. Sig. (2-tailed) | 0.194 | 0.000 * | 0.809 | 0.029 * | 0.000 * | 0.140 | 0.063 | 0.000 * |

* Significant at $\alpha = 5\%$.

Another way that COVID-19 affects food security is through income losses and demand shocks [26]. This phenomenon also occurs among Indonesian coffee farmers. According to the survey, farmers' opinions of COVID-19 impacts on their households indicate that income has decreased, although consumption, food quality, food type consumed, internet, and electricity spending have remained steady. Food prices are perceived differently by Arabica and Robusta farmers, with Arabica farmers reporting an increase and Robusta farmers reporting no change (Table 4).

**Table 4.** The proportion of coffee farmers who experienced changes as a result of COVID-19 (in percentage).

| Aspects | Arabica Farmers | | | Robusta Farmers | | |
|---|---|---|---|---|---|---|
| | No Change | Increase | Decrease | No Change | Increase | Decrease |
| income | 39.24 | - | 60.76 | 47.62 | 1.59 | 50.79 |
| expenditure | 51.90 | 35.44 | 12.66 | 80.16 | 6.35 | 13.49 |
| consumption | 75.95 | 6.33 | 17.72 | 78.57 | 0.79 | 20.63 |
| food quality | 78.48 | 2.53 | 18.99 | 57.14 | 33.33 | 9.52 |
| food price | 32.91 | 56.96 | 10.13 | 79.37 | 17.46 | 3.17 |
| food type | 81.01 | 2.53 | 16.46 | 87.30 | 8.73 | 3.97 |
| communication cost | 68.35 | 26.58 | 5.06 | 53.97 | 42.06 | 3.97 |
| electricity cost | 83.54 | 3.80 | 12.66 | 46.03 | 15.08 | 38.89 |

The Per Capita Gross Regional Domestic Product at 2010 Constant Market Prices (2018) at West Java was IDR 29.160.060, while, at Bengkulu, was IDR 22.494.840 [22]. According to farmers' assessments of income variations caused by the pandemic, incomes tend to decline in Arabica farmers (60.76%) and Robusta farmers (50.79%). According to survey data, most Arabica farmers have a constant perception of household expenditure (51.90%) followed by a decrease (35.44%). In contrast, Robusta farmers have a relatively constant perception of household expenditure (80.16%), followed by a decline (13.49%). The expenditures of Robusta farmers tended to be related to a higher proportion of food expenses.

The pandemic has varying effects on farmer households regarding food quality and pricing. The majority of Arabica and Robusta farmers have assessed the food quality to be relatively stable (57.14% and 74.48%, respectively), with increasing (33.33%) for Robusta farmers and declining (18.99%) for Arabica farmers. Arabica farmers are most affected by the increase in food prices (56.98%), while the prices remain relatively stable for Robusta farmers (79.37%).

The internet and electricity have become essential facilities during a pandemic. In 2020, the percentage of households in Bengkulu that used the internet in the last three months was 71.69%, while, in West Java, was 82.18% [22]. According to the survey data, the internet usage of Arabica farmers in West Java Province increased by 26.58% and 42.06% for Robusta farmers in Bengkulu Province. Most coffee farmers in Bengkulu and West Java considered the electricity costs relatively constant (83.54% and 46.03%, respectively), followed by a decline (38.89% and 12.66%).

### 3.2. Coffee Farming System and Impact of COVID-19

The description of the coffee farming system can be seen from several aspects, including experience, area, selling coffee type, and agriculture diversification options (Table 5). The average experience of respondent Robusta farmers was 14.09 years, compared to 8.78 years for Arabica farmers. The average Robusta coffee farmer's plantation (1.59 ha) is larger than the average Arabica coffee farmer's plantation (0.83 ha), with the smallest area 100 m². In West Java, 100% Arabica coffee is grown, with the majority (97.5%) sold as cherries, while 100% Robusta coffee is grown in Bengkulu and sold as beans (91.3%).

**Table 5.** Coffee farming system descriptions.

| Aspect | Min | Max | Mean | Std. Deviation | Description |
|---|---|---|---|---|---|
| **Arabica farmers** | | | | | |
| experience | 1 | 20 | 8.78 | 4.92 | |
| area | 0.10 | 5.00 | 0.84 | 0.78 | |
| selling coffee type | 1 | 2 | 1.03 | 0.16 | 1 = Cherry (97.5%), 2 = Beans (2.5%) |
| Mixed Crop System | 0 | 1 | 0.87 | 0.34 | 1 = Yes (87.3%), 0 = No (12.7%) |
| Crops–Livestock | 0 | 1 | 0.82 | 0.38 | 1 = Yes (82.3%), 0 = No (17.7%) |
| **Robusta farmers** | | | | | |
| experience | 1 | 55 | 14.09 | 9.42 | |
| area | 0.33 | 5.00 | 1.59 | 0.93 | |
| selling coffee type | 1 | 2 | 1.91 | 0.28 | 1 = Cherry (8.7%), 2 = Beans (91.3%) |
| Mixed Crop System | 0 | 1 | 0.18 | 0.39 | 1 = Yes (18.3%), 0 = No (81.7%) |
| Crops–Livestock | 0 | 1 | 0.21 | 0.41 | 1 = Yes (20.6%), 0 = No (79.4%) |

In terms of the diversification of farmers' activities, Arabica farmers implemented a mixed crops system (87.3%) and coffee livestock integration (82.3%) in contrast to the Robusta farmers, who integrate only 18.3% of their crops and 20.6% of their livestock. The establishment of coffee plantations for Arabica farmers is motivated by conserving plants on forest land. Farmers can continue by using the Mixed Crop System during the transition period, which entails planting horticulture between stands until the main crop produces. Additionally, the government assists farmers with livestock to mitigate environmental damage caused by the extensive use of chemical fertilizers and pesticides in previous horticultural cultivation. The purpose of this program is to promote the use of organic fertilizers. Avoiding the use of chemical fertilizers, herbicides and pesticides can also reduce carbon, nitrogen and water footprints [27].

The pandemic caused by COVID-19 significantly impacted the agriculture and the food supply chain [24]. In terms of the food system, a pandemic affects agricultural input and output markets, food processing, and the food supply chain [28]. The impact analysis in this study was conducted in greater detail in terms of input, cultivation, output, and strategy. Fertilizers, organic fertilizers, pesticides, non-family labor, agricultural tools, and working a capital comprise the input side analysis. The Mann–Whitney analysis indicates significant differences in the use of fertilizers, organic fertilizers, and pesticides among the respondent groups of Arabica and Robusta farmers (Table 6).

**Table 6.** Mann–Whitney Analysis for the COVID-19 impact on input aspect.

| | Chemical Fertilizer | Organic Fertilizer | Pesticide | External Worker | Equipment | Capital |
|---|---|---|---|---|---|---|
| Mann–Whitney $U$ | 2767.500 | 3820.500 | 3849.500 | 4751.500 | 4915.500 | 4729.000 |
| Wilcoxon W | 5927.500 | 6980.500 | 7009.500 | 12,752.500 | 12,916.500 | 7889.000 |
| Z | −5.769 | −3.558 | −3.621 | −0.715 | −0.208 | −0.645 |
| Asymp. Sig. (2-tailed) | 0.000 * | 0.000 * | 0.000 * | 0.475 | 0.835 | 0.519 |

* Significant at $\alpha = 5\%$.

Farmers' perceptions about providing chemical fertilizers, organic fertilizers, and pesticides are different. The impact of the pandemic on the provision of chemical fertilizers to Arabica farmers is very influential (51.9%), while it is somewhat influential (60.32%) on Robusta farmers (Figure 3). According to Arabica farmers, the impact of the pandemic on the supply of organic fertilizers is very influential (64.56%), followed by extremely influential (25.32%). In contrast, for Robusta farmers, it is very influential (76.98%) and followed by the somewhat influential category (17.46%). Arabica farmers use chemical fertilizers, organic fertilizers, and pesticides, but fertilizers and chemical pesticides have not been used in conjunction with balanced management, while, for Robusta farmers, very small use does not cause any disturbance in procurement.

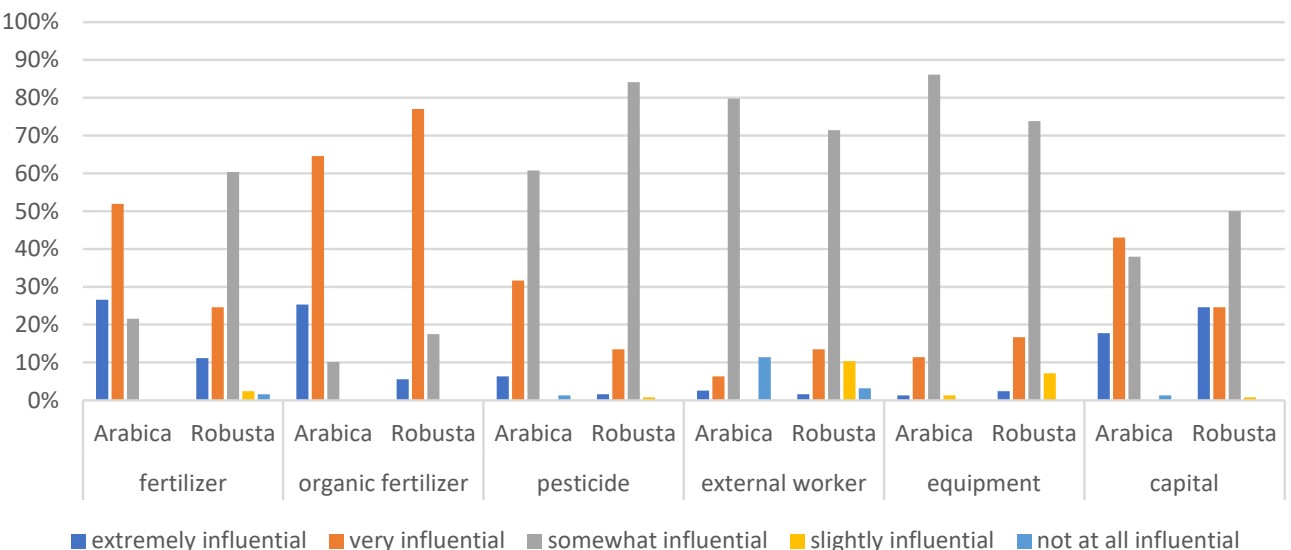

**Figure 3.** The proportion of coffee farmers who experienced affects in the farming input aspect as a result of COVID-19.

The majority of Robusta farmers did not fertilize their crops. In Bengkulu, low-income areas, the relatively high cost of fertilizers discourages fertilizing. Organic fertilizers made from animal or plantation waste are likewise not produced. Apart from their lack of animals, farmers have historically avoided using compost made from coffee husk waste. The challenges associated with integrating coffee livestock as a source of organic fertilizer are related to production input, labor, expertise, technology, money, and farmer institution [29]. As a result, Robusta farmers have a moderate perception of the pandemic's impact on input components. In contrast, Arabica farmers believe that COVID-19 significantly impacts the input aspect.

In general, the GAP application for Robusta smallholders is still limited. Low productivity is related to seedling problems and a lack of GAP due to farmers' limited capital to apply coffee cultivation technology by technology recommendations from seedling, cultivation, harvest, and post-harvest.

The production and the price at the farm level will determine the farmer's income. GAP application will affect the production related to fertilizers, pesticides, water management, soil management, and control of pests and plant diseases. At the same time, farm-level prices vary according to the variety, quality, and market target [4]. Specifically, the impact of a pandemic on coffee management was seen by applying cultivation and the pest disease management, harvesting method, selling method, and transaction method. Analysis with the Mann–Whitney shows differences in the transaction method (Table 7).

**Table 7.** Mann–Whitney analysis for the COVID-19 effects on the process aspect of coffee farming systems.

| | Cultivation | Pest Disease Management | Harvesting | Selling Method | Transaction Method |
|---|---|---|---|---|---|
| Mann–Whitney *U* | 4594.500 | 4615.500 | 4614.500 | 4973.500 | 4429.500 |
| Wilcoxon W | 7754.500 | 7775.500 | 12,615.500 | 12,974.500 | 12,430.500 |
| Z | −1.287 | −1.197 | −1.324 | −0.011 | −2.047 |
| Asymp. Sig. (2-tailed) | 0.198 | 0.231 | 0.185 | 0.991 | 0.041 * |

* Significant at $\alpha = 5\%$.

COVID-19 has a very influential impact on GAP adoption (87.34%) and pest disease management (86.08%) among Arabica farmers (Figure 4). Implementing crop management is more challenging during a pandemic due to cost allocation and societal constraints affecting the availability of agriculture workers. Since cultivation activities have not been carried out intensively according to the GAP, this is not a significant barrier for Robusta farmers. The difference is in the mode of transaction, which Robusta farmers find to be more challenging. This situation is intrinsically linked to Arabica farmers' more significant communication technology and infrastructure advantage.

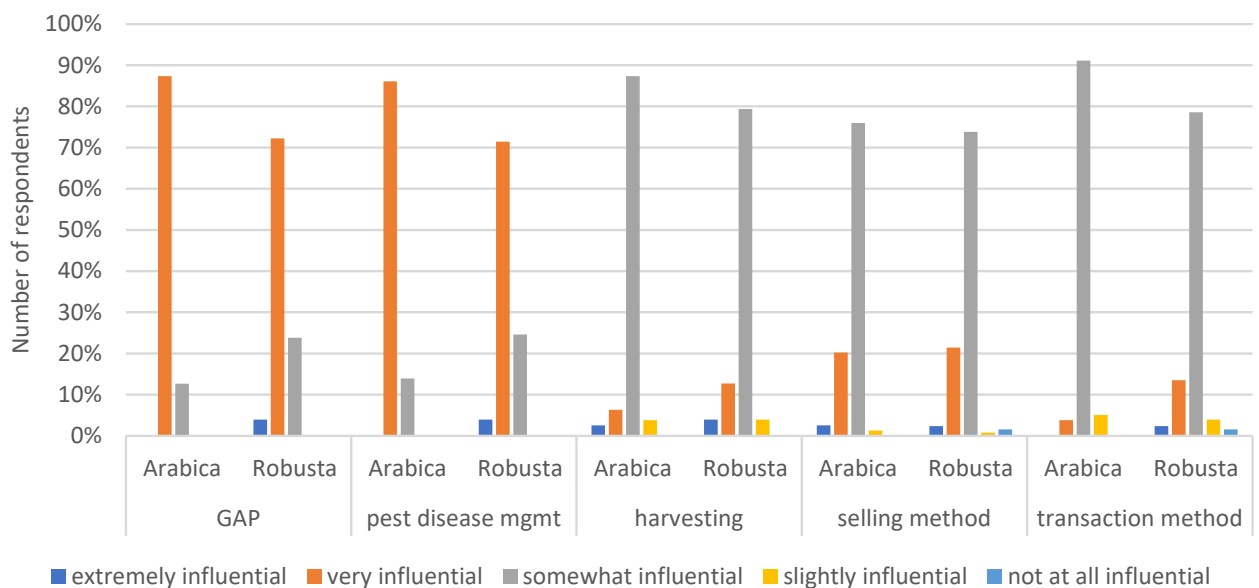

**Figure 4.** The proportion of coffee farmers who experienced affects in the farming process aspect as a result of COVID-19.

Arabica farmers harvest red picks, while most Robusta farmers implement random picking. Both Arabica and Robusta farmers have not changed their harvesting and picking procedures. The pandemic does not affect this condition. Red picking is more often associated with specialty coffee, and it is necessary for Arabica coffee. Another issue is that Robusta farmers tend to be at risk of coffee theft, so harvesting is often done before all red fruit has ripened.

From a marketing perspective, online sales through e-commerce platforms are more of an option than offline sales, which are currently difficult. However, not all farmers, particularly Robusta farmers, have expertise in selling online. As a result, Robusta farmers are more affected by the pandemic's influence on transaction techniques.

*3.3. Innovation Sharing Model Development to Support Coping COVID-19 Strategy Implementation*

The pandemic has had a relatively significant impact on agricultural inputs, particularly chemical fertilizers for Robusta farmers and organic fertilizers for Arabica and Robusta farmers. The cause of this condition has external and internal aspects. The external aspect is concerned with implementing restrictions on community activities, which impacts the distribution of agricultural inputs. The internal aspect has to do with farmers' declining economic capacity, which will impact their ability to purchase fertilizer.

On the other hand, when faced with shocks that affect the ability of farmers to break even, they reduce or forgo important but expensive agricultural maintenance activities, resulting in decreased quality and yields and, some, even substitute commodities [30]. In some areas in Indonesia, even before COVID-19, coffee trees may have been unable to reach their full potential due to the soil's nutrient balance being disturbed by the lower fertilizer rates [5]. Coffee quality is influenced by GAP and Good Manufacturing Practices (GMP). These procedures ensure the fulfillment of food safety requirements, ranging from production conditions to processing and storage facilities and personal hygiene [31].

Given that the pandemic's duration cannot be predicted, coping strategies are needed to overcome the decline in the use of inputs in smallholder coffee plantations. Based on the previous analysis, it is known that not many farmers own livestock, while, based on farmers' perceptions, COVID-19 impacts the procurement of fertilizers and the application of GAP, including fertilization. Therefore the Integrated Crops Livestock System (ICLS) is a recommended risk management approach [32]. This approach aligns with the recommendation that using more livestock for income stability is one of the coping strategies [10]. In addition, the implementation of ICLS provides benefits in reducing inorganic fertilizers, pesticides, and other inputs [33].

Organic matter application, including green manure and organic manure, has been found to increase soil organic matter by partially supplying plant nutrient requirements and promoting the growth of soil microbes [34]. In addition to increasing organic matter, organic amendments can also increase soil organic carbon and crop yields [35].

Fertilization with conservation can improve the soil quality of coffee plants, allowing them to perform better, have a higher growth score, greater resistance to pests and diseases, and increase production [36]. The scarcity of inorganic fertilizers due to distribution disruptions during the pandemic can be overcome using organic fertilizers. However, the use of organic fertilizers is also inhibited by scarcity at the farmer's level [37]. Adopting the best local fertilizer management practices is a strategic step in nutrient management and the sustainability of coffee production [5].

Coffee livestock integration has several benefits, including enhanced coffee plant productivity, increased animal weight, decreased fertilization costs, decreased livestock management costs, increased soil fertility, and an improved land structure to prevent land erosion [38]. Compost and urine nutrients can considerably boost the coffee yield, while organic fertilizers can assist plants in preparing for production [39]. Farmers will be able to satisfy their fertilizer demands due to the deployment of an integrated system and better risk management through diversification of agriculture activities.

Adopting ICLS as a coping strategy required investigating the farmers' capacity and ability, which necessitated cluster analysis. Each cluster's features are determined by the values of each variable (Table 8), and the distances between the final cluster centers are provided in Table 9. Cluster 1 is defined by using organic fertilizers, livestock ownership, animal waste as fertilizer use, difficulties in providing forage for livestock, pealing coffee cherries, and the use of coffee waste for fertilizer and feeds through the processing phase. Cluster 2 is the opposite of cluster 1 in that all aspects are not carried out. Cluster 3 is defined by using organic fertilizers, livestock ownership, animal waste as fertilizer, and difficulties in providing forage for livestock; nevertheless, coffee plantations and livestock waste are not used for feed and fertilizer. Cluster 4 is defined by using organic fertilizers,

ownership of livestock, animal waste as fertilizer, barriers to livestock forage fulfillment, and animal waste as organic fertilizer.

**Table 8.** Description of the final cluster.

| | Cluster | | | |
|---|---|---|---|---|
| | **1** | **2** | **3** | **4** |
| use organic fertilizer | 1 | 0 | 1 | 1 |
| livestock farming | 1 | 0 | 1 | 1 |
| use of animal manure as fertilizer | 1 | 0 | 1 | 1 |
| facing limited forage fodder | 1 | 0 | 1 | 1 |
| peeling coffee cherries | 1 | 0 | 0 | 1 |
| use coffee waste for fertilizer | 1 | 0 | 0 | 1 |
| processing fertilizer from coffee waste | 1 | 0 | 0 | 1 |
| use coffee waste for feed | 1 | 0 | 0 | 0 |
| processing feed from coffee waste | 1 | 0 | 0 | 0 |

Description: 1 = yes, 0 = no.

**Table 9.** Distances between final cluster centers.

| **Cluster** | **1** | **2** | **3** | **4** |
|---|---|---|---|---|
| 1 | - | 2.571 | 1.908 | 1.267 |
| 2 | 2.571 | - | 1.358 | 2.117 |
| 3 | 1.908 | 1.358 | - | 1.525 |
| 4 | 1.267 | 2.117 | 1.525 | - |

The ANOVA table indicates which variable had the most significant influence on the COVID-19 coping strategy in each cluster. Variables with high F values offer the most separation between clusters. According to the ANOVA results, all variables have a significance value of less than 0.05. (Table 10). Since it has the highest F value, the variable "peeling coffee cherries" is the one that most distinguishes cluster members.

**Table 10.** Analysis of Variance.

| **Aspects** | **Cluster** | | **Error** | | **F** | **Sig.** |
|---|---|---|---|---|---|---|
| | **Mean Square** | **df** | **Mean Square** | **df** | | |
| use organic fertilizer | 8.892 | 3 | 0.120 | 201 | 74.057 | 0.000 |
| livestock farming | 13.044 | 3 | 0.056 | 201 | 233.486 | 0.000 |
| use of animal manure as fertilizer | 7.768 | 3 | 0.089 | 201 | 87.478 | 0.000 |
| facing limited forage fodder | 9.458 | 3 | 0.081 | 201 | 116.068 | 0.000 |
| peeling coffee cherries | 14.353 | 3 | 0.005 | 201 | 3000.280 | 0.000 |
| use coffee waste for fertilizer | 13.683 | 3 | 0.009 | 201 | 1527.927 | 0.000 |
| processing fertilizer from coffee waste | 4.238 | 3 | 0.078 | 201 | 54.436 | 0.000 |
| use coffee waste to feed production | 2.506 | 3 | 0.027 | 201 | 91.130 | 0.000 |
| processing feed from coffee waste | 5.962 | 3 | 0.005 | 201 | 1246.383 | 0.000 |

According to the findings, as many as 84.13% of Robusta farmers exclusively cultivate coffee with low technology and do not have livestock, implying any coffee livestock integration (Figure 5) due to farmers' limited knowledge and access to information technology and their limited skills. Farmers with only a primary education account for a sizable proportion

of the sample at 24.6%, and farmers range from 20 to 73 years. A similar pattern occurs among Arabica farmers; even though the majority of them already have livestock, just 21.52% of them integrate coffee livestock into a closed cycle. Another 65.42% of farmers have not taken advantage of the mutual relationship between crops and livestock to improve productivity and profitability due to farmers' limited capital.

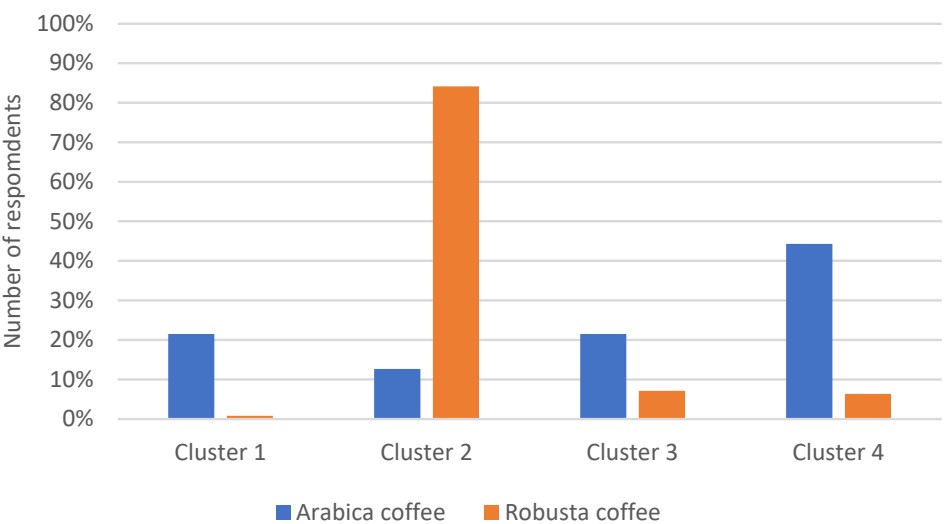

**Figure 5.** Number of respondents in each cluster on coffee type based.

Based on the results, the strategy that must be developed is to accelerate the coffee livestock integration as a coping strategy for COVID-19 for smallholders. Due to the increased management complexity of ICLS and incentives or disincentives for ICLS adoption in a given policy environment, socioeconomic factors will affect the benefits that can be realized beyond crop yields alone, production, and farmers' incomes [32]. Due to the variability of the current conditions, technical strategies must be followed by formulating specific institutional strategies. Agricultural technology adoption positively correlates with farmer education, household size, land size, credit availability, land tenure, extension services access, and organization membership [40]. For some farmers, the main barrier is a lack of expertise, but for others, the technology cannot be implemented due to a lack of capital.

The adoption of coffee livestock integration results in various additional operating costs, the majority of which are based on the geographic proximity of the farms [41]. This barrier, however, can be solved by increasing the scope of integration in terms of coverage area and participant count. Implementing ICLS in market-oriented farms provides a strategy for increasing the probability of ICLS success [42]. Third-party coordination is critical, as is financial and technical support.

Agricultural innovations are primarily about increasing the production of food, fodder, or secondary products and enhancing the quality of produce, production process, or growing conditions [43]. Given that technology adoption is a dynamic process rather than a one-time static decision, it is directly linked to knowledge collecting, learning, and experience [44]. More flexible and adaptable technology transfer initiatives that address the demands of resource-poor smallholder farmers are needed [45]. Consequently, it is necessary to develop a model that can clearly distinguish these problems as elements of innovation and aspects of the innovation process (Figure 6).

The proposed model of innovation sharing elements and the process for smallholders are separated into two components: innovation sharing elements concerned with inputs and resources and the innovation sharing process involved with the process of technology dissemination and adoption. Farmers, content, and technology all contribute to the innovation sharing elements. Farmer resources are categorized according to their asset,

capacity, and capability. The term content refers to material for innovation in the form of data, information, and knowledge.

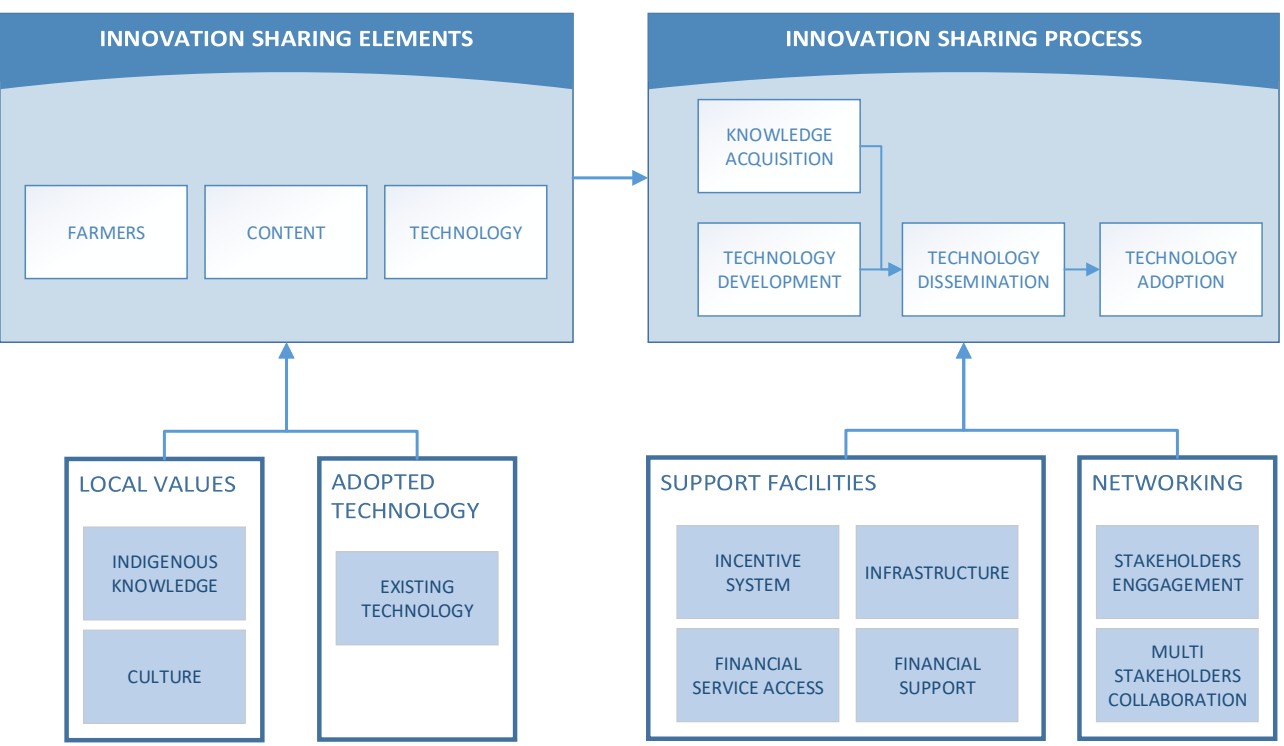

**Figure 6.** Proposed model of innovation sharing for smallholders.

The term technology refers to agricultural technology developed to increase productivity and overcome chemical, physical, biological, and social constraints inherent in crop production systems [46]. Agriculture in the Industry 4.0 Era will involve supply chain transformation and adaptation to digital change, making it necessary to examine the starting points and existing conditions [47]. The local form of knowledge expands in the context of the area, and specific features are shaped by the culture and economy [48]. In addition to cultural factors, the success of agricultural knowledge management operations is influenced by institutional, individual, and knowledge factors. As a result, parts of innovation sharing will function if indigenous knowledge, culture, and existing technology are considered.

The process of innovation sharing includes knowledge acquisition, technology development, dissemination, and adoption. Knowledge acquisition is the process of gathering or collecting knowledge from various sources. Agriculture in the future will rely heavily on advanced technologies like robotics, temperature and moisture sensors, aerial imagery, and GPS. In the coffee sector, the Internet of Things, machine learning, and geostatistics are the most often used technology [49]. Sustainable practices in the coffee sector can be improved by digital quality management, advanced process control, and statistical process control tools [50]. Farms will be able to operate more profitably, efficiently, safely, and sustainably with the support of this sophisticated technology, precision agriculture, and robotic systems. While advanced technology and cutting-edge innovation promise huge benefits, access to practical technology for smallholders is promoted to maximize their land and labor potential. As a result, the path of technology development is toward adapting technology to specific location requirements.

A well-designed combination of technology dissemination approaches would be most effective and have the most potential for long-term adoption [51], which support facilities and networking influence the innovation sharing process. Support facilities consist of

an incentive system, infrastructure, financial service, and financial support. Networking consists of stakeholders' engagement and stakeholder collaboration. Partnerships for data innovation integrate agricultural data from field, laboratory, and greenhouse studies through various sensors, tools, and apps and provide a rapid display and summary of statistics for real-time decision-making [52].

Based on the proposed model of innovation sharing for smallholders, the strategy for cluster 1 is directed at providing support facilities and networking development to optimize innovation sharing towards sustainable adoption at the farmer level. The strategy includes capacity development for farmers and farmer organizations. Capacity development fosters entrepreneurship to strengthen group-based firms and create group-level business models to gain economies of scale.

The strategy for cluster 2 is to identify indigenous knowledge and existing technology, prepare farmers' conditions, develop knowledge management, and increase access to technology sources. The strategy includes providing knowledge and technology linked to enhancing GAP implementation and livestock farming systems and integrating them at the individual farmer level. Extension initiatives will be critical.

Clusters 3 and 4 focus on the innovation sharing process to encourage technology adoption on a wider spectrum. Specifically, Cluster 3 suggests increasing the added value of coffee and reusing waste. It is vital to transfer information and offer equipment and machines to farmers in the early phases. Farmers are expected to have knowledge and skills connected to coffee processing and waste management for fertilizer and animal feed. Farmers who previously sold coffee in the form of cherries have moved their focus to sell coffee in the form of beans.

Cluster 4 proposes expanding coffee waste, which can be used as fertilizer and feed to alleviate farmers' fodder scarcity. The utilization of coffee waste for feed is guided by a processing method that results in higher quality and productivity. Additionally, it is vital to map out government initiatives or chances for collaboration in cattle procurement in this cluster.

Cluster development in implementing coffee livestock integration relates to integration, including farm-to-farm partnerships, farmer groups, and regional integration [53]. The first and most basic form is a farm-to-farm partnership, in which specialized crop and livestock farms exchange raw materials (manure, grain, fodder, and straw). The second type of direct exchange can be organized by the local crop and livestock farmers negotiating land use allocation patterns. A third type involves upscaling to a regional scale, where spatially separated groups of specialized livestock and crop farmers integrate through third-party coordination such as agricultural cooperatives.

## 4. Conclusions

COVID-19 impacts Arabica and Robusta farmers, both on farmer households and the coffee farming system. The impact is not always directly on productivity and quality but is more typically generated by disruptions in the production system. Based on coffee varieties Arabica and Robusta, differences occurred in the intensity of the impact, which is inseparable from the farming system implemented before the pandemic, where Arabica farmers carried out more intensive cultivation.

From the farmer's household side, their income decreased significantly, affecting the intensity of crop management. Meanwhile, in the farming system, this impact is mainly on the procurement of inputs for chemical fertilizers and organic fertilizers and farmers' decisions to fertilize.

The appropriate coping strategy is the application of coffee livestock integration. Due to the variability of current conditions, the strategy developed should be cluster-specific. Interventions are carried out differently in each cluster after the model of innovation sharing for smallholders is developed. The proposed model is separated into two components: innovation sharing elements concerned with inputs and resources, and the innovation sharing process concerned with the process of technology dissemination and adoption.

When farmers have previously established coffee plantations and livestock management, the program is directed at providing support facilities and networking development to optimize innovation sharing towards sustainable adoption. Suppose farmers have not yet implemented coffee livestock integration, the intervention is directed at identifying indigenous knowledge and existing technology, preparing farmers' conditions, developing knowledge management, and increasing access to technology sources. The intervention will focus on information and technology transfers related to increasing the GAP, livestock farming methods, and individual farmer-level integration. When integration cannot be accomplished in a closed system, it focuses on the innovation sharing process to encourage technology adoption in a broader spectrum. The cluster description will promote the process of extension strategies and policies aimed at certain farmer groups.

**Author Contributions:** Conceptualization, S.W. and F.D.; methodology, S.W. and R.V.; formal analysis, S.W. and R.V.; resources, F.D.; writing—original draft preparation, S.W., F.D. and R.V.; writing—review and editing, S.W. and R.V.; supervision, F.D.; and funding acquisition, F.D. All authors have read and agreed to the published version of the manuscript.

**Funding:** This work was funded, in part, by the Australian Center for International Agricultural Research (ACIAR) through their Meryl Williams Fellowship program.

**Acknowledgments:** We would like to express our deep gratitude to John Gibson and Rebecca Spence from the University of New England, Australia, for their patient guidance and enthusiastic encouragement of this work.

**Conflicts of Interest:** The authors declare no conflict of interest.

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
