# Peer review of "Coping Strategies of Smallholder Coffee Farmers under the COVID-19 Impact in Indonesia"

_agriculture, doi:10.3390/agriculture12050690_

Round 1
Reviewer 1 Report
Reviewer’s report
Specific comments
Line 1: Title is little unorganized. Instead it may be put as “Coping strategies of smallholder coffee farmers under the Covid-19 impacts in Indonesia”
Line 9-24: Abstract need to be improved. It should begin with real problem faced by the smallholder coffee farmers in the region followed by study aims. Analysis used ……….analysis (line 12-13) should be better expressed in terms of use of statistical tools for exploring the facts. Line (16-17)… “Coffee livestock integration” is a strategy to improve farmers’ livelihood to mitigate the impact. This line suggests that it is already an established strategy, however, authors have explored it as one of the important strategy to compact the negative impact. Thus, the sentence may be improved accordingly, like “it was explored by the study that Coffee livestock integration is an important option for enhancing livelihoods of farmers as well as to mitigate the impact of Covid-19.”
Line 27-99: Under the introduction, the status of coffee production and its stakeholders need to be explained so as to justify the importance of study. How much of coffee production in Indonesia by smallholders is contributing to the national economy/GDP? These facts and figures need to be explained with latest datasets.
Line 107-119: Geographical location of study area need to be portrayed by means of map.
Line 116-117: Months of study should be indicated apart from a small mention of climate of the study area (rainfall, min and max temp, season, radiation, etc). This type of mention will improve the application of study in similar agro-ecologies.
Author Response
Dear Reviewer
We would like to express our gratitude to the reviewers for their thorough and insightful review of our manuscript. We appreciate your precious time in reviewing our paper and providing valuable comments. It was your valuable and insightful comments that led to possible improvements in the current version. We have considered the comments and tried our best to address every one of them. Attached we provide the point-by-point responses.
Regards,
Authors

Reviewer 2 Report
Review on agriculture-1698882 -Impact of Covid-19 and coping strategies of smallholder Coffee farmers in Indonesia
1) Lines 88-92: These information should not be here. It would be better to mention about these before you stated the research questions/ research objectives.
2) Lines 116-119: The criteria for selecting the respondents must be mentioned. Who should be selected and interviewed? It must be based on some potential of farmers such as farm experience, be the farm owner, farm size, …so on. All coffee farmers cannot randomly selected, if your random got a new farmer with 0.6 to 1 year experience for coffee cultivation, he/she may not be the good respondent.
3) Line 142: How to obtain the variables in Table 1? Are they suggested by the previous studies? Providing the references are useful.
4) Statistic analysis: Please mention about testing the significant difference of variables between Arabica farmers and Robusta farmers.
5) Lines 171-189: Are there any significant difference of household characteristics between Arabica farmers and Robusta farmers?
6) Lines 192-197: Covid-19 pandemic is also impacting farmer’ household food security.
7) Line 200: Why Arabica and Robusta farmers’ perceptions are quite different? The explanations will be useful.
8) Lines 251-254: Avoiding the use of chemical fertilizers, herbicides and pesticides can also reduce carbon, nitrogen and water footprints.
9) It would be clearer and easy to understand if the results in Tables 7-9 can be presented in the Graph forms.
10) Lines 350-352: Applying organic amendment can not only increase organic matter, but can increase soil organic carbon and crop yield.
11) Line 379: The result in Table 10 should be presented in the Graph form.
12) Line 419: “(Jabbar et al., 2003)” The reference’ format must be revised.
Author Response
Dear Reviewer
We would like to express our gratitude to the reviewers for their thorough and insightful review of our manuscript. We appreciate your precious time in reviewing our paper and providing valuable comments. It was your valuable and insightful comments that led to possible improvements in the current version. We have considered the comments and tried our best to address every one of them. Attached we provide the point-by-point responses.
Regards
Authors

Round 2
Reviewer 2 Report
Accept in present form.